# Predictive Biomarkers for Checkpoint Inhibitor-Based Immunotherapy: The Galectin-3 Signature in NSCLCs

**DOI:** 10.3390/ijms20071607

**Published:** 2019-03-31

**Authors:** Carlo Capalbo, Giorgia Scafetta, Marco Filetti, Paolo Marchetti, Armando Bartolazzi

**Affiliations:** 1Department of Medical Oncology, Sant’Andrea University Hospital, 00189 Rome, Italy; carlo.capalbo@uniroma1.it (C.C.); marco.filetti@uniroma1.it (M.F.); paolo.marchetti@uniroma1.it (P.M.); 2Department of Molecular Medicine Sapienza University of Rome viale Regina Elena 324, 00161 Rome, Italy; 3Department of Oncology-Pathology Sant’Andrea University Hospital, via di Grottarossa 1035, 00189 Rome, Italy; giorge@hotmail.com; 4Department of Oncology-Pathology, Cancer Center Karolinska, Karolinska Hospital, S-17176 Stockholm, Sweden

**Keywords:** galectin-3, non-small cell lung carcinoma, pembrolizumab, checkpoint inhibitors, predictive marker

## Abstract

Checkpoint inhibitor-based immunotherapy is opening a promising scenario in oncology, with objective responses registered in multiple cancer types. However, reliable predictive markers of tumor responsiveness are still lacking. These markers need to be urgently identified for a better selection of patients that can be candidates for immunotherapy. In this pilot study, a cohort of 34 consecutive patients bearing programmed death-ligand 1 (PD-L1)-positive non-small cell lung carcinoma (NSCLC), treated with pembrolizumab, was considered. The retrospective immuno-phenotypic analysis performed on the original tumor biopsies allowed for the identification of a specific “galectin signature”, which strongly correlated with tumor responsiveness to anti PD-1 immunotherapy. We observed that the large majority of patients (about 90%) with high galectin-3 tumor expression (score 3+) showed an early and dramatic progression of the disease after three cycles of treatments. In contrast, all patients with negative or low/intermediate expression of galectin-3 in tumor cells showed an early and durable objective response to pembrolizumab, indicating galectin-3 as an interesting predictive marker of tumor responsiveness. The galectin-3 signature, at least in NSCLCs, promises a better selection of patient candidates for immunotherapy, reducing unnecessary treatment exposures and social costs. A large multicenter study is ongoing to validate this finding.

## 1. Introduction

Checkpoint inhibitor-based immunotherapy is going to alter the approach to cancer treatment. A meaningful improvement in survival outcomes versus standard of care has been reported in monotherapy as well as in combined therapeutic approaches [1,2,3,4]. Several monoclonal antibodies to programmed cell death protein-1 (PD-1) and programmed cell death protein-1 ligand (PD-L1) have been approved for clinical use. Although it is irrefutable that checkpoint inhibitors-based immunotherapy is opening a new era in oncology, important clinical and economical aspects have to be considered. In particular, it is urgent to optimize the clinical use of these tools by improving the selection of potentially responsive tumors. This will reduce patients’ exposure to ineffective and potentially harmful treatments, as well as social costs.

It has been recently documented in a growing number of malignancies that tumor mutation burden (TMB) and microsatellite instability status are more informative biomarkers for predicting the response to anti-PD-L1 immunotherapy than PD-L1 immunohistochemical determination [5,6,7]. In particular, although a companion immunohistochemical method for PD-L1 expression analysis has been approved by the U.S. Food and Drug Administration (FDA) and the European Medicines Agency (EMA) for patients with advanced non-small cell lung carcinomas (NSCLC) as a specific requirement for treatment with pembrolizumab, the predictive value of this preliminary assay remains questionable [4,5]. Indeed, a significant percentage of PD-L1-positive NSCLCs cases do not respond to anti-PD-L1 or anti-PD-1 immunotherapy. On the other side, a significant number of PD-1-negative tumors are sensitive to this therapy [4,5,8].

Tumor cells–immune system interactions, as well as interactions of cancer cells with extracellular matrix and host tissues, involve complex and dynamic mechanisms evolving from the initial malignant transformation of a normal cell to tumor growth and progression. Immune evasion is the constant feature in the journey of cancer cells, and understanding the molecular mechanisms regulating this function will allow the adoption of effective therapeutic strategies. As a general concept, the clinical applicability of the different strategies of cancer immunotherapy requires an improved understanding of the biological mechanisms limiting these approaches [9].

Recently, the biological and predictive roles of PD-L1 were analyzed together with some functional aspects of potential predictive biomarkers, which are, likely, key determinants for tumor responsiveness to checkpoint inhibitor-based immunotherapy [5]. In this context, the tumor microenvironment (TME) and tumor infiltrating lymphocytes play a critical role. In particular, CD8+ T cell density, CD8+ T cells/CD3+ FoxP3+ regulatory T cell ratio [5,6], T cell receptor (TCR) clonality [6], mutational or neoantigen burden [10,11,12], and IFNγ-related immune signature were studied to predict responsiveness to checkpoint inhibitor-based immunotherapy [13,14,15].

Actually, the efficacy of anti-PD-L1 or anti-PD-1 immunotherapy is limited by several and complex immunosuppressive mechanisms, which are present, at least in part, within the TME [5]. In this context, important regulatory mechanisms inducing immunosuppression occur through galectins [16,17].

Galectins, a family of evolutionary conserved animal lectins that bind N-acetyllactosamine, play different biological functions in cancer, inflammation, and immune tolerance [16,17]. These molecules can be expressed in tumor cells and in the extracellular milieu and are virtually expressed, constitutively or transiently, in immune cells and tumor-infiltrating lymphocytes. Galectins bind multiple glycosylated structures on the cell surface as well as in the extracellular matrix and are in the right position for translating glycan-encoded information into specific biological programs with relevant functional implications in immune cell activation, differentiation, homeostasis, tumor survival, and tumor immune escape [16,17,18,19,20,21,22]. Specific galectins have been reported to bind also glycosylated ligands on the cell surface of the host immune cells, modulating their functions [17]. Galectins also bind sugar residues of cytokines, allowing their functional inactivation [17,21].

In this complex scenario, galectin-1 and galectin-3 deserve consideration for the functional implications they can have on tumor immune response. Expression of galectin-1 and galectin-3 in tumor cells can block apoptosis; when shed in the tumor microenvironment, both galectins induce T cell apoptosis via CD45 and CD7 binding on T cell surface, favoring, indeed, tumor immune escape [17].

These specific galectins also bind the TCR, contributing to altered signaling and activation of T lymphocytes. In particular, galectin-3 multivalent N-glycan complex impairs TCR clustering on T cell surface, increasing the agonist threshold for TCR signaling [17]. Changes in glycosylation of cell surface glycoproteins can selectively control the survival of T_H_1 and T_H_17 lymphocytes, modulating their susceptibility to galectin-1-induced apoptosis [19]. The molecular cross-talk between T cell surface glycosylation and some galectins is, indeed, functionally active in modulating T lymphocytes death and inflammatory response [17,18,19,20,21]. Antigen presenting cells (APCs) and macrophages play also an important role in establishing immune cell homeostasis. Their dual role in orchestrating protective immune responses or immune tolerance is likely due to specific galectin–glycan interactions [5]. Marked glycan changes have been discovered during dendritic cell maturation, modulating the binding of specific galectins to mature or immature APCs [5].

Galectin-3 is expressed in a consistent percentage of NSCLCs, as well as in other malignancies. Soluble galectin-3 derived by tumor shedding, binds specific glycan residues in the tumor microenvironment, forming a complex lattice. This complex molecular structure reduces IFNγ diffusion through the tumor matrix and the chemokine gradient that is necessary to favor T lymphocytes migration into the tumor [21]. The finding that experimental transfer of cytotoxic T lymphocytes in vivo reduces tumor growth only after galectin-3 inactivation further support the key role of galectin-3 in favoring tumor immune escape [23]. This scenario opens the possibility that a tumor-specific “galectin signature” could be a surrogate predictive marker of tumor responsiveness to checkpoint inhibitors-based immunotherapy. To investigate this hypothesis, the expression profile of galectin-3 was assessed in a panel of PD-L1-positive (tumor proportion score > 50%) non-oncogene-addicted treatment-naive metastatic NSCLCs treated with pembrolizumab.

## 2. Results and Discussion

Thirty-four consecutive patients with PD-L1-positive NSCLCs stage IV, treatment-naive, were considered for pembrolizumab (200 mg intravenously every 3 weeks) as first-line treatment. The characteristics of these patients are summarized in Table 1. The expression analysis of galectin 3 was performed retrospectively on tumor biopsies at the time of diagnosis (Figure 1). Tumors were grouped in different classes depending on the expression level of galectin-3 as assessed by immunohistochemistry (IHC) (Table 1).

In total, 21 out of 34 patients (62%) had high galectin-3-expressing tumors, whereas 13 out of 34 (38%) showed negative–low/intermediate galectin-3 expression. Almost all (19 out of 21) patients belonging to the high galectin-3 expression group (scored 3+) showed a dramatic progression of the disease after three cycles of treatments, despite high PD-L1 expression. Interestingly, in two of these instances with a homogeneous expression of galectin-3 (> 95% of positive cells), an early hyper-progressive disease was observed. Tumor resistance to pembrolizumab strongly correlated with high galectin-3 expression (p: 0.0001) (Appendix A). Conversely, the majority of patients with negative–low/intermediate expression of galectin-3 in tumor cells had an early and durable objective response to pembrolizumab. In particular, patients with galectin-3-negative or heterogeneous tumor expression showed the best therapeutic response according to immune-related response criteria (iRECIST) (Table 1 and Figure 2). Very interestingly, the only patient with negative/low galectin-3 expression, who had not responded after 12 weeks of treatment, showed a delayed partial response after 24 weeks (patient n. 21 in Figure 2). Finally, squamous histology seemed to correlate with low and intermediate expression of galectin-3, although this finding needs to be confirmed because of the low number of cases analyzed.

Regarding treatment-related toxicity, no grade 3–4 adverse events were observed.

Predictive markers of tumor responsiveness to immunotherapy are part of a complex and interconnected multifactorial scenario (tumor–host microenvironment), in which checkpoint inhibitors play key roles. The pleiotropic molecule galectin-3 regulates apoptosis and tumor immune escape and may play a role as a predictive marker of NSCLC responsiveness [16,17,21]. These data, together with the experimental evidence of the biological effects of galectins in tumor growth, progression, and immune escape, provide the rationale for “looking in the sugar box” in order to identify a tumor-specific galectin signature [24,25]. In particular, this pilot study discovered that tumor resistance to pembrolizumab strongly correlated with high expression of galectin-3 in cancer cells, whereas an objective clinical response was obtained in the vast majority of tumors with a negative–low or intermediate galectin-3 phenotype after 12 and 24 weeks of treatment. This clinical observation is supported by a strong biological rationale. Galectin-3 has been demonstrated to play a key role in regulating the immune response [17]; indeed, galectin-3 secreted by different tumor cells has been shown to change macrophage polarization from M1, the anti-tumor macrophage, to M2 [17,20]; trigger CD8 T-cell apoptosis and restrict T-cell receptor (TCR) clustering [17,18,19]. Moreover, galectin-3 is able to capture IFNγ in the tumor matrix, reducing chemokine gradient production and T cell tumor infiltration [21]. All together, these biological effects contribute to cancer immune escape [17,21,22,25,26]. To date, several galectin-3 inhibitors (i.e., GR-MD-02 and GCS-100) are under clinical investigation both alone and in combination with check-point inhibitors in different cancer settings [27].

The major limitations of the present pilot study are clearly represented by the small number of patients considered and by the relatively short follow-up. This weakness, however, does not minimize the biological value of the message arising from this study: (i) the expression of PD-L1 or PD-1 is not sufficient per se to predict tumor response to pembrolizumab, (ii) the galectin-3 signature, at least in NSCLCs, has the potential to be a surrogate marker of tumor responsiveness to pembrolizumab. A large multicentric study involving four independent lung cancer institutions at the national level is in progress with the aim to validate our findings.

If the results of this pilot study are confirmed, a cheap and easy immunohistochemical method could be used to evaluate the expression of galectin-3 in tumor biopsies, allowing a better selection of patient candidates for immunotherapy. This approach will avoid exposure to ineffective treatments as well as social costs.

Specific studies to better understand if this new and clinically relevant function of galectin-3 is restricted to lung carcinomas or represents, instead, a general mechanism of cancer cell resistance to immunotherapy are strongly encouraged and may provide an important contribution to cancer therapy and glycobiology as well.

## 3. Methods

### 3.1. Patients

Thirty-four consecutive patients with NSCLC stage IV, treatment-naive, selected on the basis of PD-L1 tumor proportion score (TPS) ≥ 50%, non-oncogene-addicted were considered for immunotherapy with pembrolizumab (200 mg intravenously every 3 weeks) as a first-line treatment. The majority of patients had a diagnosis of adenocarcinoma (28/34), two of which with a solid pattern of growth at histology; the remaining six tumors were classified as primary squamous cell carcinomas. Patients assessments were conducted at baseline and after three and six cycles of therapy, and patients were classified according to iRECIST.

### 3.2. Monoclonal Antibodies and Immunohistochemistry

Mouse monoclonal antibodies to PD-L1 (clone 22C3 DakoCytomation, Glostrup, Denmark) and a rat mAb to Galectin-3 (clone M3/38, Mabtech, Nacka, Sweden) were commercially acquired and used according to the manufacturer’s instructions. Briefly, antigen-retrieval microwave treatment (0.01M citrate buffer pH 6.0) was applied when required for three cycles of 5 min each, at 750 W. Endogenous peroxidase activity was quenched with methanol–hydrogen peroxide (3%) for 15 min. After blocking with unrelated antiserum, tissue slides were incubated with the primary monoclonal antibodies in a moist chamber at 4 °C. The immune reaction was visualized by using the Envision System (DakoCytomation, Glostrup, Denmark) for indirect immunoperoxidase, as required.

The expression analysis of PD-L1 and galectin-3 was performed by IHC on NSCLC tumor biopsies collected before treatment. Microscopic evaluation was performed independently by two experienced pathologists. Two discordant IHC evaluations were resolved after a consensus meeting with experts in the pathology laboratory. Most importantly, galectin-3 expression analysis was performed in blind, meaning that no clinical information was available to pathologists at the time of IHC evaluation of the tumors. PD-L1 over-expression in tumor cells (>50%) was considered imperative in order to start immunotherapy with pembrolizumab. The expression analysis of galectin-3 in our cohort of NSCLCs allowed to identify three groups of patients: (a) a high galectin-3 expression group (score 3+) with positive tumor cells >70%; (b) an intermediate galectin-3 expression group (scores 2+ and 1+) with 50–70% and 10–50% positive cells, respectively; (c) a low expression group (negative score and +/− score) including galectin-3-negative cases and cases showing scattered positive tumor cells <10% [23]. Tissues labelled without the primary antibody were used as negative controls. Tumor biopsies were collected at Sant’Andrea University Hospital and used for immunohistochemical assays in full agreement with the guidelines provided by the Institutional Review Board (Prot. CE no. 8391/2013) and Helsinki Declaration. A written informed consent was obtained from all patients enrolled in the study.

### 3.3. Molecular Analysis

All tumors treated with pembrolizumab were also assessed for the presence of *EGFR* gene mutations (exons 18-19-20-21), ALK translocation, or ROS1 rearrangement by using molecular biology techniques, FISH analysis, and immunohistochemistry as appropriate. All tumors tested were negative for known oncogenic driver mutations (Appendix A).

### 3.4. Statistical Analysis

Fisher’s exact test was used in the analysis of contingency table (Appendix A).

## Figures and Tables

**Figure 1 ijms-20-01607-f001:**
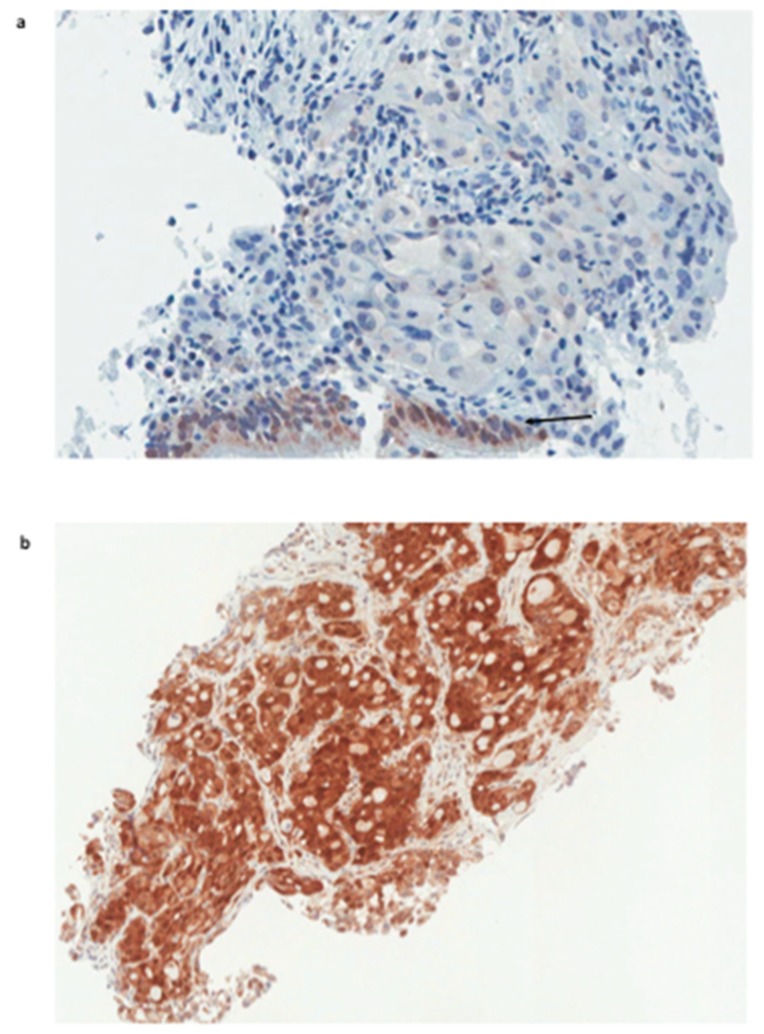
Galectin-3 expression in NSCLC biopsies. (**a**) Squamous cell carcinoma, galectin-3-negative. Note the expected expression of galectin-3 in normal bronchial epithelium at the bottom of the tumor, which represents the internal positive control (arrow). (**b**) High galectin-3-positive lung adeno-carcinoma (score 3+). (Magnification 250×; Immunoperoxidase staining as reported in the material and method section).

**Figure 2 ijms-20-01607-f002:**
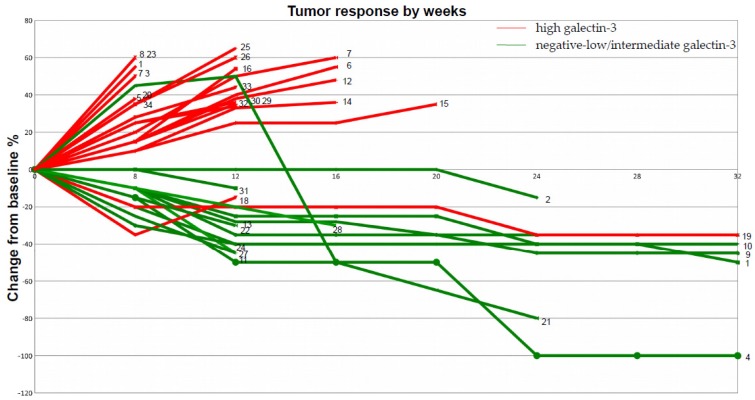
Spider plot of tumor response to checkpoint inhibitor therapy. The clinical response of each patient listed in Table 1 is plotted in this picture to better correlate galectin-3 expression with tumor sensitivity to pembrolizumab during follow-up. According to Table 1: patients with high galectin-3 expression (red lines), the large majority of which showed disease progression after 20 weeks of follow-up; patients with negative–low/intermediate galectin-3-expressing tumors (green lines) showing significant sensitivity to pembrolizumab. One patient had a complete response (n°4);

**Table 1 ijms-20-01607-t001:** Clinical features of the patients and non-small cell lung carcinoma (NSCLC) phenotypes.

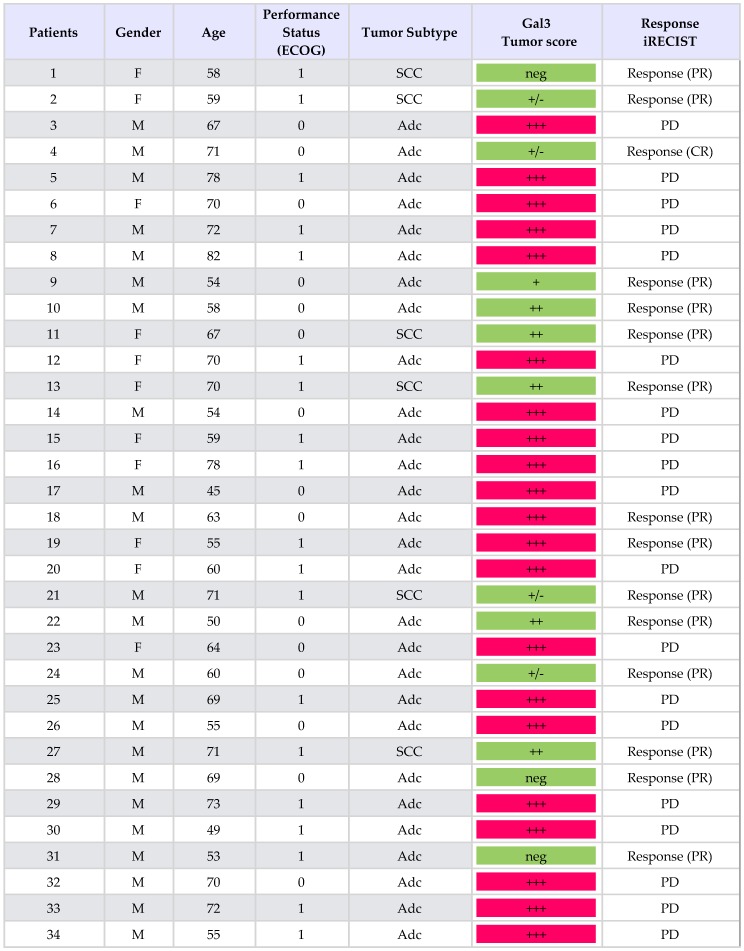

Legend: F: Female; M: Male; SCC: Squamous Cell Carcinoma; Adc: Adenocarcinoma; Gal3: Galectin-3 expression (neg: negative; +/−: < 10%; +: >10<50%; ++: >50<70%; +++: >70%). Immune-related response criteria (iRECIST): CR: Complete Response; PR: Partial Response; SD: Stable Disease; Progressive Disease: PD.

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
