# Peer review of "Predictive Biomarkers for Checkpoint Inhibitor-Based Immunotherapy: The Galectin-3 Signature in NSCLCs"

_ijms, 2019, doi:10.3390/ijms20071607_

Reviewer 1 Report

The manuscript by Capalbo et all studies the predictive value of galectin signature on the responsiveness of non-small cell lung carcinoma to PD-L immunotherapy using pembrolizumab. In 34 analyzed samples monitored for up to 32 weeks, the authors found that tumor resistance to pembrolizumab strongly correlated with high galectin-3 expression in the tumor tissue as determined immunohistochemically. I find the manuscript scientifically sound and the results  interesting to the readers of the Journal. However, the presentation of the results and the technical level of the manuscript should be improved. I recommend the manuscript for publication after addressing the points below.

- I do not find the title justified. The whole work is concentrated on studying galectin-3 expression and its correlation to the tumor response to treatment. There is no experimental examination of carbohydrates involved. I suggest to include the work "galectin-3" or ""galectin signature" instead of "sugar" into the title.

- The manuscript is not organized according to the Instructions for Authors (neither the correct order of Sections nor their titles). There is no conclusion in the manuscript. The references are not formatted according to the Instructions for Authors and there are no doi as required. Please address these points in the revision.

- I recommend to simplify the last column in Table 1, cancelling the cathegory "stable disease". It was found only in two patients (2, 31) and in both there was response to treatment observed during the monitored time, though somewhat delayed. Personally, I would just enter cathegory "response" instead of SD, PR and CR and add a clarifying footnote to the one patient with complere response (4). Thus, the Table would become clearer, easier to orient in and more connected to Fig 2 where there are only two colors anyway. Moreover, I suggest to make the Table directly in the text processor (and not as embedded figure) for better legibility.

- Fig 2 - the title, axis descriptions, and legends are too small and badly legible.

- l. 184-185. The authors state that there are studies on galectin-3 inhibitors in combination with check-point inhibitors but give no reference. Could you please add the respective reference(s)?

And, last but not least, could the authors add the information on the levels of galectin-3 expression in the serum of the studied patients? Galectin-3 was classified as an official biomarker connected to heart disorders but sofar, not in cancer. It would be very inspiring to add the results on the serum blood levels of galectin-3 since they are much easier to monitor and it may reasonably be supposed that they would correlate with the the immunohistological results.

Author Response

Reviewer 1:

We thank this reviewer for the useful suggestions

The manuscript has been improved according to the reviewer’s suggestion.

1.     The title has been definitively changed as required

2.     ……The manuscript is not organized according to the Instructions for Authors (neither the correct order of Sections nor their titles). There is no conclusion in the manuscript. The references are not formatted according to the Instructions……

- This manuscript is not a full paper but it should be considered a brief comunication reporting fresh original clinical relevant data, which need to trigger a multicenter study in short time. 

- In the revised version Results and Conclusions are merged and this specific section has been improved.

- References have been extensively revised as required. New pertinent references are also added in the revised version.

      3.  Last column of table 1 has been rewritten according to the reviewer’s suggestion.

In this format connection with figure 2 is definitively easier.

(We provide the revised table 1 separately in order to facilitate the editor to process it).

4.     Figure 2 has been improved as suggested.

5.     The requested reference has been provided (ref. 26)

6.     …..could the authors add the information on the levels of galectin-3 expression in the serum…….?

It is well-known that soluble galectin-3 could be a potential marker for different diseases, including cancer. Galectin-3 has been extensively used as predictive marker implicated in cardiac fibrosis and heart failure. However many inflammatory conditions may also contribute to increase the serum levels of galectin-3.

Although this represents an interesting and promising field of research to be pursued, it is necessary to study the normal serum level of galectin-3 in “normal matching (age, sex, life-style, etc…) subjects” in order to define the ideal cut-off level. After that a large screening of patients bearing lung carcinoma (different stages); pneumonia; and other lung diseases will be necessary to validate the possibility to use galectin-3 as soluble diagnostic or predictive marker of lung carcinoma. This comparative study requires a different work-hypothesis and resources, but deserves of course to be considered in the next future.

Reviewer 2 Report

Cancer immunotherapy with immune checkpoint inhibitors is currently the standard for the treatment of many cancers. Unfortunately, we still do not know in which patients this form of treatment will be successful. Predictor factors are still searched. In spite of a small group, the authors proved that galectin-3 could be such a factor. This interesting discovery requires further research on a larger group of patients, to which I encourage authors. I highly recommend this article for publication in “International Journal of Molecular Sciences”. My suggestions. - Citations in line 68 and 80 are missing - please unify the spelling of galectin-3 (gal-3 or Gal-3?) - in my copy the figures and tables are blurred.

Author Response

Reviewer 2:

We thank this reviewer for the useful suggestions and his/her kind consideration about this paper

-       A multicenter study at National level is ongoing.

-       The missing references have been provided.

Reviewer 3 Report

The manuscript by Capalbo and colleagues is interesting in that it reports galectin-3 expression as a predictor of response to PD-1 blockade in NSCLC patients. While the findings seem logical and supported by the data, the samples size, analysis, and limitations of the study are very concerning. However, the report could be useful to clinicians and researchers trying to identify prognostic markers for potential lung cancer patients targeted for treatment with immunotherapies. I envision that with some major revisions, the manuscript could be accepted into IJMS. Please see the specific comments below:

1.       The grammar and punctuation need to be polished for the next revision:

For example:

 Line 13: “several tumors” to “various cancers” or “multiple cancer types”

Line 15: “In this study,”

Line 18: “allowed for the identification of”

Line 30: “change consistently cancer therapy” to “alter the approach to cancer treatment” etc.

Line 34: “Although it is…”

Line 73: “altered signaling”

Line 108: “reaction” is misspelled

Line 137: “belonging”

Line 178: “last” to “vast”

There are numerous grammatical issues present in the manuscript that need to be corrected before acceptance.

2.       Line 45: is there a reference pertaining to IHC having questionable predictive power?

3.       Line 47: There is also a significant portion of PD-1 negative tumors that respond to therapy. Perhaps this should be discussed in the context of this study.

4.       It may be worth mentioning other pro-cancer immune cells such as MDSCs or alternate macrophages – these are mentioned slightly in the discussion but not in the intro. Galectins also play a key role in mediating monocyte function, yet, this is not mentioned.

5.       Line 58: There are multiple mechanisms for immunosuppression (i.e. TGF-beta). I would at least mention the possibility that other mechanisms may influence immunosuppression in NSCLC.

6.       Do you have staging criteria for each patient? Pertinent demographics from the patients are missing and may provide evidence of confounding, effect modification, etc.

7.       I find quite a few mentions of “data not shown” that could support the claims of the manuscript. Perhaps it would be beneficial to add these as supplementary files?

8.       Was any quality control used for the FFPE staining for PD-L1? I do not see a control cell line (i.e. MCF-7) or slides without secondary antibody. Also, a positive control (NCI-H226) would have been appropriate to ensure the staining protocol was accurate and reproducible throughout several sections of the same block of tissue.

9.       Line 111: Why were the pathologists not blinded for evaluation of IHC slides? This could introduce bias to scoring slides in both PD-L1 expression and galectin-3 expression.

10.   Line 126 – this data would be beneficial as a supplementary file. Tumors harboring genetic mutations could influence response to immunotherapies, thus, any differences in mutational burden could not only influence response, but also galectin-3 expression.

11.   All SCC patients responded to therapy. Perhaps you could stratify you analyses by cancer subtype. Do the results from your test still hold true when you remove SCC cases from adenocarcinomas? Additionally, only three patients scored negative for galectin, I worry about such a small sample size (hence the Fishers test instead of chi squared). Were these patients combined with the low/intermediate galectin samples for the analysis? Since you have the age of each patient, why was analysis on age not performed as it could potentially confound your effect? At first glance, it seems that the younger patients responded better to therapy. (Table 1)

12.   Only one complete response? Can you speculate as to why most patients only showed a partial response to therapy?

13.   Line 140: were these the only patients to have >95% galectin staining?

14.   Figure 1 – perhaps mention the score for the SCC sample in a. If possible, show controls for your staining.

15.   Figure 2: This graph is difficult to read, and it looks as if several patient’s data are censored. Were they lost to disease, failure to follow up, etc? I feel that a survival analysis should have been performed and would make a much better argument for your hypothesis. Additionally, you might want to perform a Cox regression and report the appropriate hazard ration for each level of galectin staining.

16.   The limitations of the study are not addressed in the discussion and need to be presented. A study with such a small sample size has several.

17.   Line 179 – 183 This sentence needs reformatting. Also, where are the references for these mechanisms? It is generally appropriate to reference the original paper along with the review article when citing previous findings (ref 23).

18.   As noted previously, I have strong concerns over the limitations of this study. Further evidence using a larger sample size would need to be presented to support the claims made in the manuscript. Additional information on inclusion/exclusion criteria for such a study would also be appropriate.

Author Response

       Reviewer 3:

We would like to thank this competent reviewer for the constructive and pertinent suggestions. First of all it is necessary to stress that we presented here the results of a pilot study. The manuscript indeed is not a full paper but it should be considered a brief comunication reporting fresh original clinically relevant data, which should trigger  multicenter studies in short time. 

1.     We did our best to improve grammar and punctuaction. Hopefully, all the changes required have been applied.

2.     The required references have been provided (ref. 4, 5, 8)

3.     A specific sentence (information) has been added in the introduction section

4.     A specific mention on galectins and APCs /macrophages function has been added to the introduction as required.

5.      A general overview on mechanisms that may influence immunosuppression in cancer has been  considered in the revised manuscript.

6.     All the patients enrolled in this study are stage IV patients, treatment-naive, selected on the basis of PD-L1 tumor proportion score (TPS) ≥ 50 % and bearing non-oncogene-addicted NSCLCs. In table 1 are reported the patients’ features. A supplementary table 1 showing the molecular features of the tumors has been provided.

7.     It has been done

8.     The use of adequate negative controls for IHC has been finally specified in the revised material and methods section.

This work has been performed in a pathology department of St. Andrea Hospital in which IHC is routinely used for diagnosis. The lab has a very high experience not only in this technique but also in production and characterization of mAbs for diagnostic use. (This can be easily verified following the scientific production of the senior author on PubMed). Several mAbs produced and characterized in our lab are used worldwide. Of course all the immunohistochemical procedures performed in the lab have adequate negative and positive controls. IHC for PD-L1 is reported as row data in the supplementary table 1. IHC  for galectin-3 is shown in figure 1. In panel a) the internal positive control for galectin-3 immunostaining is represented by the normal ciliated bronchial epithelium on the bottom of the figure (arrow) to demonstrate the quality of the immune-staining. Panel b) show 100% positive adenocarcinoma (negative control was prepared incubating the tissue section with unrelated mAb or omitting the first mAb. Data not shown.  The suggested use of slides without secondary antibody as negative control is not appropriate because in this case the chromogenic reaction cannot be triggered in indirect immunoperoxidase assay ! ).

9.     This specific point has been clarified in the revised material and method section.

10.  A supplementary table1 has been provided as suggested

11.  The suggestions reported in this point will be fully considered for the analysis of data deriving by the ongoing multicenter study involving more than 150 cases. The limited number of cases analyzed in the present pilot study and the limited number of SCC in our cohort do not allow further statistical considerations.

12.  See point 11.

13.  There were several patients with homogeneous galectin-3 expression in the cluster of 3+ score.

14.  A positive internal control (arrow) was applied. The staining is clearly negative for the tumor (a SCC)

15.  Figure 2 has been improved. The survival analysis was not the scope of this pilot study but it will be considered in a retrospective multicenter study (in progress) with adequate long follow-up. At this time the study was mostly focused to analyze the presence of clinical objective response to cancer immunotherapy.

16.  Limitations of the study have been adressed in the discussion section.

17.  The suggested changes have been applied and some references have been added.

18.  As aforementioned a large multicenter study is ongoing. This brief comunication is instrumental for triggering different multicenric studies in the field of cancer immunotherapy to be included in the new special issue of IJMS entitled:  Galectins in Cancer and translational medicine 2.0.  The revised version of this paper, if suitable for publication, will represent the first original contribution .

Round  2

Reviewer 3 Report

The grammatical and spelling errors have yet to be rectified, and these issues would need to be resolved before publication. However, the results and conclusions drawn from these results seem appropriate and well founded. Perhaps the biggest issue with the study is the lack of sample size. Assuming a larger study is indeed in the works, the conclusions drawn from this very preliminary study may help better predict response to PD-1 inhibition. With some clarity and revisions to the below mentioned issues, I would recommend this article for publication pending minor revisions.

There are still several grammatical errors within the manuscript that I would strongly encourage the authors to resolve before publication:

Line 45 – “exposure”

Line 46 – as well as social costs.

Line 52 – the FDA and the EMA

Line 53 – as a specific requirement for treatment with

Line 62 – resulting or regulating?

Line 70 – in this context, the TME

Line 73 - ,and IFN

Line 76 – where are the references supporting these mechanisms?

Line 89 – remove – “it is relevant the concept”

Line 93 – consideration for what?

Line 93- Expression of Galectin-1 and Galectin-3 in tumor cells – instead of “their”

Line 99 – the T cell surface

Line 145 – be consistent with capitalization of Galectin or galectin

Line 159 – Tissues labelled without primary antibody were used as negative controls.

2.       It should be noted that the driver mutations that were tested is not a comprehensive list (lung cancers often have oncogenic mutations or amplifications of several other RTKs)

Table/Figure 1 – should be capitalized throughout the manuscript

Supplemental Table/Figure – should also be capitalized

Line 178 – Expression analysis of galectin – remove “as aforementioned”

Line 180 – what marker? “grouped into categories based on galectin-1 expression as assessed by IHC”

Line 186 – interestingly,

Line 190 – Patient 21 did not have an early response, thus, this statement is inaccurate

Line 195 – Patient 21 did NOT have a complete response as shown in Table 1; it tells us PR

Line 197 – although,

3.       There is no description of Table 1 in the Figure legend only abbreviations 

Line 208 - Figure 1 legend – Galectin-3

Line 216 – red patients does not describe the group – replace with “patients with high galectin expression (red line)” – similarly, “green patients” is not grammatically corrected

Line 221 – tumor-host

Line 222 – play key roles.

Line 224 – may play a role as a predictive marker…

Line 228 – this pilot study discovered

Line 233 – this clinical observation is supported by a strong

Line 233 – remove “in fact”

Line 235 – 240 – it would be clearer if each reference was inserted after the described mechanisms

Line 242 – settings

Line 248 – a surrogate

Line 249 – multicentric?

Line 250 – institutions at the national level

Line 251 – were confirmed,

Line 254 – exposure to ineffective treatments as well as social costs、

Author Response

Minor changes required

First of all we would like to thank the reviewer for the useful advices provided for  improve the manuscript.

There are still several grammatical errors within the manuscript that I would strongly encourage the authors to resolve before publication:

All the following changes have been applied as requested

Line 45 – “exposure”

Line 46 – as well as social costs.

Line 52 – the FDA and the EMA

Line 53 – as a specific requirement for treatment with

Line 62 – resulting or regulating?

Line 70 – in this context, the TME

Line 73 - ,and IFN

Line 76 – where are the references supporting these mechanisms?

Line 89 – remove – “it is relevant the concept”

Line 93 – consideration for what?

Line 93- Expression of Galectin-1 and Galectin-3 in tumor cells – instead of “their”

Line 99 – the T cell surface

Line 145 – be consistent with capitalization of Galectin or galectin

Line 159 – Tissues labelled without primary antibody were used as negative controls.

2.       It should be noted that the driver mutations that were tested is not a comprehensive list (lung cancers often have oncogenic mutations or amplifications of several other RTKs).

Although many other genetic alterations can be considered in the NSCLC, Pembrolizumab as monotherapy is indicated for the first-line treatment of metastatic non-small cell lung carcinoma (NSCLC) in adults whose tumours express PD-L1 with a ≥ 50% tumour proportion score (TPS) with no sensitizing EGFR mutations or ALK translocations. In the present study we considered the minimal genetic alterations necessary for the first-line immunotherapic treatment as indicated by regulatory agencies (FDA and EMA). In particular, this patients selection is supported by the phase III KEYNOTE-024 registrative trial, in which 305 treatment-naïve patients with advanced NSCLC having at least 50 percent tumor cell PD-L1 staining were randomly assigned to pembrolizumab monotherapy versus standard platinum-doublet chemotherapy

Reck M, Rodríguez-Abreu D, Robinson AG, Hui R, Csőszi T, Fülöp A, Gottfried M, Peled N, Tafreshi A, Cuffe S, O'Brien M, Rao S, Hotta K, Leiby MA, Lubiniecki GM, Shentu Y, Rangwala R, Brahmer JR, KEYNOTE-024 Investigators N Engl J Med. 2016;375(19):1823. Epub 2016 Oct 8.

…….Table/Figure 1 – should be capitalized throughout the manuscript

Supplemental Table/Figure – should also be capitalized

The requested changes have been applied

Line 178 – Expression analysis of galectin – remove “as aforementioned”

This correction has been applied

Line 180 – what marker? “grouped into categories based on galectin-1 expression as assessed by IHC”

Line 186 – interestingly,

These corrections have been applied

Line 190 – Patient 21 did not have an early response, thus, this statement is inaccurate

This specific point has been corrected

Line 195 – Patient 21 did NOT have a complete response as shown in Table 1; it tells us PR

This error  has been corrected

Line 197 – although,

This change It has been applied

3.       There is no description of Table 1 in the Figure legend only abbreviations

A title/description  for table 1 has been provided by Editor on the top of the table

Line 208 - Figure 1 legend – Galectin-3

Line 216 – red patients does not describe the group – replace with “patients with high galectin expression (red line)” – similarly, “green patients” is not grammatically corrected

The requested changes have been applied

Line 221 – tumor-host

Line 222 – play key roles.

Line 224 – may play a role as a predictive marker…

Line 228 – this pilot study discovered

Line 233 – this clinical observation is supported by a strong

Line 233 – remove “in fact”

Line 235 – 240 – it would be clearer if each reference was inserted after the described mechanisms

Line 242 – settings

Line 248 – a surrogate

Line 249 – multicentric?

Line 250 – institutions at the national level

Line 251 – were confirmed,

Line 254 – exposure to ineffective treatments as well as social costs

The changes listed above have been applied